# The Free Myocutaneous Tensor Fasciae Latae Flap—A Workhorse Flap for Sternal Defect Reconstruction: A Single-Center Experience

**DOI:** 10.3390/jpm12030427

**Published:** 2022-03-09

**Authors:** Amir Khosrow Bigdeli, Florian Falkner, Benjamin Thomas, Gabriel Hundeshagen, Simon Andreas Mayer, Eva-Maria Risse, Leila Harhaus, Emre Gazyakan, Ulrich Kneser, Christian Andreas Radu

**Affiliations:** Department of Hand, Plastic and Reconstructive Surgery, Burn Center, BG Trauma Center Ludwigshafen, Plastic and Hand Surgery, University of Heidelberg, Ludwig-Guttmann-Str. 13, D-67071 Ludwigshafen, Germany; amir.bigdeli@bgu-ludwigshafen.de (A.K.B.); florian.falkner@bgu-ludwigshafen.de (F.F.); benjamin.thomas@bgu-ludwigshafen.de (B.T.); gabriel.hundeshagen@bgu-ludwigshafen.de (G.H.); simon.mayer@bgu-ludwigshafen.de (S.A.M.); eva-maria.risse@bgu-ludwigshafen.de (E.-M.R.); leila.harhaus@bgu-ludwigshafen.de (L.H.); emre.gazyakan@bgu-ludwigshafen.de (E.G.); ulrich.kneser@bgu-ludwigshafen.de (U.K.)

**Keywords:** sternal defect reconstruction, deep sternal wound infection, DSWI, reconstructive microsurgery, free flap, tensor fasciae latae flap, TFL flap

## Abstract

Introduction: Deep sternal wound infections (DSWI) after cardiac surgery pose a significant challenge in reconstructive surgery. In this context, free flaps represent well-established options. The objective of this study was to investigate the clinical outcome after free myocutaneous tensor fasciae latae (TFL) flap reconstruction of sternal defects, with a special focus on surgical complications and donor-site morbidity. Methods: A retrospective chart review focused on patient demographics, operative details, and postoperative complications. Follow-up reexaminations included assessments of the range of motion and muscle strength at the donor-site. Patients completed the Quality of Life 36-item Short Form Health Survey (SF-36) as well as the Lower Extremity Functional Scale (LEFS) questionnaire and evaluated aesthetic and functional outcomes on a 6-point Likert scale. The Vancouver Scar Scale (VSS) and the Patient and Observer Scar Assessment Scales (POSAS) were used to rate scar appearance. Results: A total of 46 patients (mean age: 67 ± 11 years) underwent sternal defect reconstruction with free TFL flaps between January 2010 and March 2021. The mean defect size was 194 ± 43 cm^2^. The mean operation time was 387 ± 120 min with a flap ischemia time of 63 ± 16 min. Acute microvascular complications due to flap pedicle thromboses occurred in three patients (7%). All flaps could be salvaged without complete flap loss. Partial flap loss of the distal TFL portion was observed in three patients (7%). All three patients required additional reconstruction with pedicled or local flaps. Upon follow-up, the range of motion (hip joint extension/flexion (*p* = 0.73), abduction/adduction (*p* = 0.29), and internal/external rotation (*p* = 0.07)) and muscle strength at the donor-sites did not differ from the contralateral sides (*p* = 0.25). Patient assessments of aesthetic and functional outcomes, as well as the median SF-36 (physical component summary (44, range of 33 to 57)) and LEFS (54, range if 35 to 65), showed good results with respect to patient comorbidities. The median VSS (3, range of 2 to 7) and POSAS (24, range of 18 to 34) showed satisfactory scar quality and scar appearance. Conclusion: The free TFL flap is a reliable, effective, and, therefore, valuable option for the reconstruction of extensive sternal defects in critically ill patients suffering from DSWIs. In addition, the TFL flap shows satisfactory functional and aesthetic results at the donor-site.

## 1. Introduction

Deep sternal wound infection (DSWI) after cardiac surgery is a severe complication that can result in devastating mortality rates between 10 and 47% [1,2]. Among these multimorbid patients, who often not only bear the burden of pre-existing coronary artery disease (CAD), but also chronic obstructive pulmonary disease (COPD) and diabetes mellitus (DM), reconstructive procedures are highly challenging [2]. The treatment of DSWIs requires radical surgical debridement of soft and bony tissue and, eventually, tension-free defect closure with well-perfused tissue [3]. For sternal defect reconstruction, microsurgeons have a vast armamentarium of pedicled and free flaps at their disposal: Common reconstructive options for sternal defect reconstruction are local flaps from the chest, abdomen, or the back, such as the vertical rectus abdominis musculocutaneous (VRAM) flap, the pectoralis major, and the latissimus dorsi (LD) flap [4,5]. When both internal mammary arteries (IMAs) have been harvested for coronary-artery bypass grafts (CABG) or after previous local flap failure, reconstruction can be difficult. This is partially due to the fact that the arc of rotation of pedicled flaps is limited [5,6] and closure of defects, which include the entirety of the sternum, can be critical, putting the most distal part of the pedicled flap at risk of impaired perfusion [7,8]. To offer these multimorbid patients the best possible care and optimal long-term outcomes, we are increasingly using the free myocutaneous tensor fasciae latae (TFL) flap for extended deep sternal defect reconstruction. With its voluminous muscle bulk and large skin paddle, it can be adapted to large and multilayered wounds, making it ideal for complex sternal reconstruction [9,10]. Here, we report our one-decade single-center experience of 46 free TFL flaps for deep sternal defect reconstruction following cardiac surgery. The study aimed at evaluating the feasibility of this free flap for sternal reconstruction by analyzing operative data, surgical complication rates, reconstructive outcomes, and donor-site morbidity.

## 2. Patients and Methods

The study has been performed in accordance with the guidelines and regulations of the Declaration of Helsinki and has been approved by the local ethics committee (Mainz, Germany, IRB approval reference number: 2021–15577). Retrospective clinical data were collected from our institutional database of patients undergoing free flap reconstruction from January 2010 until March 2021. A retrospective chart review for intraoperative details, surgical and medical complications, length of hospitalization, as well as outcome and mortality rates, was performed. The severity of DSWI was rated according to the El Oakley and Wright classification from I to V [11]. The ASA (American Society of Anesthesiologists) classification system was used to assess the perioperative risk for each patient. Postoperative surgical complications, which required additional surgical intervention, were considered as major. The primary outcomes studied were “re-explorations” because of acute vascular complications, such as pedicle thromboses, as well as flap necrosis, wound dehiscence, hematoma, and infection. Partial flap necrosis was considered as necrosis affecting >5% (maximum of 20%) of the flap surface area. In addition, any medical complications, such as respiratory failure or death throughout the postoperative hospital stay, were evaluated.

### 2.1. Pre-, Intra-, and Postoperative Treatment

All free TFL flap operations were performed in a two-team approach for the donor- and recipient-sites. Intraoperatively, 500 IU to 1500 IU (international units) of unfractionated heparin were applied prior to releasing the flap anastomosis or 2000 IU to 3000 IU in case of an arteriovenous loop (AVL). Intraoperative flap perfusion measurements were not performed regularly. However, since January 2017, indocyanine green angiography (ICG) has been performed occasionally, depending on the individual intraoperative decision of the senior surgeon. Primary closure of the TFL donor-site was performed in two layers. Closed suction drains were left in situ in all cases. No additional reconstruction of the fascia was performed. Postoperatively, all patients received 30 mg enoxaparin twice a day over a five-day period, followed by daily 40 mg doses for at least two weeks. Subsequently, the therapy was continued until adequate mobilization of the patient was achieved All free flaps were examined hourly by analyzing the capillary refill, skin temperature, and skin color for five days in order to detect any perfusion alterations.

### 2.2. Follow-Up

Follow-up was established from the date of surgery to the last outpatient visit at our department at least 3 months after discharge. Follow-up examinations included donor-site range of motion (ROM), with measurements in hip and knee joints, as well as strength measurements of the thigh muscles. Results were analyzed and compared to the contralateral healthy side. Muscle strength was assessed manually and scaled in six grades (0 = complete paralysis, 1 = contraction palpable, 2 = active movement with gravity eliminated, 3 = active movement against gravity, 4 = active movement against resistance, 5 = normal power) [12]. Additionally, each patient completed the Quality of Life 36-item Short Form Health Survey (SF-36) and the Lower Extremity Functional Scale (LEFS) questionnaire [13,14]. The subjective donor-site morbidity and satisfaction with the overall aesthetic and functional results were analyzed using a self-reported non-standardized 6-point Likert-questionnaire. Results were rated on a scale from 1 to 6 (1 = excellent, 6 = poor). The Vancouver Scar Scale (VSS) and the Patient and Observer Scar Assessment Scales (POSAS) were used to analyze scarring at the donor- and recipient-sites [15,16].

### 2.3. Statistical Analysis

Data were collected in excel sheets, and statistical analyses were performed using GraphPad Prism 7.0 (GraphPad Software, Inc., San Diego, CA, USA). For descriptive statistics of patients and flaps, the mean of continuous data accompanied by the standard deviation and the median or mode with interquartile ranges for ordinal data were reported. The paired two-sided Wilcoxon–Mann–Whitney-Test and the two-sided chi-square test were computed to assess differences in categorical and dichotomous variables, respectively. *p*-values of <0.05 were regarded statistically significant.

## 3. Results

Between January 2010 and March 2021, 46 patients underwent sternal defect reconstruction with free TFL flaps. Patients included 17 women (37%) and 29 men (63%), with a mean age of 67 ± 11 years (range: 38 to 85 years). Sternal osteomyelitis in all 46 patients was confirmed upon microbiology, clinical presentation, histology, and computed tomography. The defect etiologies were as follows: wound infection and sternum osteomyelitis after CABG with use of the left internal mammary artery (LIMA, *n* = 35; 76%), CABG with use of both IMAs (*n* = 1; 2%), valve replacement (VR, *n* = 5; 11%), or combined valve replacement and LIMA-CABG (*n* = 5, 11%). The median ASA classification was 3 (range: 2 to 4). Demographic data of patients, individual risk factors, chronic conditions, the El Oakley classification, as well as the results of microbiological examinations are summarized in Table 1.

A total of 16 patients (35%) underwent secondary reconstructions at our institution following failed prior attempts. Of these, 11 patients (*n* = 11; 24%) had previously undergone unsuccessful bilateral pedicled pectoralis major flaps at the referring cardiosurgical departments. Five patients, in detail, two VRAM and three bilateral pectoralis major flaps, developed partial flap losses requiring further free flap surgery at our institution (*n* = 5; 11%). These 16 patients underwent an average of 3 ± 2 debridements and negative pressure wound therapy cycles prior to free flap surgery. The mean operation time (OT) was 387 ± 120 min (range: 212 to 695 min), which included a mean flap ischemia time of 63 ± 16 min (range: 32 to 91 min). While the mean sternal defect size was 194 ± 43 cm^2^, the mean skin paddle surface of the TFL flap was 205 ± 38 cm^2^, with a mean flap length of 24 ± 3 cm and a mean flap width of 8 ± 1 cm. Single-stage AVLs were necessary in 22 cases to provide reliable recipient vessels (Figure 1). Operative characteristics are presented in Table 2.

### 3.1. Postoperative Complications

The immediate postoperative course was uneventful in 38 of 46 patients (83%). Eight patients (17%) experienced acute microvascular complications or progredient hematoma at the recipient-site, which required emergency free flap re-exploration. In detail, acute microvascular compromise was observed during re-exploration in three cases (7%) in the form of acute arterial (*n* = 2; 4%) or venous thrombosis (*n* = 1; 2%). In this context, the use of single-stage AVLs did not increase the risk of microvascular thrombosis (*n* = 1/21 vs. *n* = 2/22; odds ratio: 0.50; 95% confidence interval: 0.03 to 4.6; *p* = 0.9). Postoperative hematoma evacuation was necessary in five patients (11%) within the first three days after flap surgery. All flaps could be salvaged. The further postoperative course was complicated in 6 patients (13%). In three patients, partial flap necroses of the most distal TFL parts were observed (7%). In these three patients, further secondary reconstructions with an intercostal anterior perforator flap (*n* = 1; 2%), a pedicled VRAM flap (*n* = 1; 2%), and two opposing local rotational flaps (*n* = 2; 4%) were performed. Wound dehiscence of the TFL flap with the need for debridement and secondary split-thickness skin-grafting (SSTG) was necessary in three patients (7%). Surgical donor-site complications occurred in five patients (11%), including three cases of impaired wound healing (7%), one case of donor-site infection (2%), and one case with the need for hematoma evacuation (2%). Of these patients, two received SSTG (4%) and three underwent successful secondary wound closure (7%). Eventually, all donor-sites healed satisfactorily. Table 3 shows the distribution of surgical complications. The average hospital stay was 34 ± 12 days, with a mean length of postoperative hospital stay of 23 ± 14 days. Postoperatively, 37 patients (80%) were monitored at the intensive care unit (ICU) for an average of 6 ± 9 days. Postoperative medical complications were seen in six patients (13%). These included: paralytic ileus (*n* = 1; 2%), postoperative delirium (*n* = 1; 2%), and cardiovascular instability with severe hypotension (*n* = 2; 4%). Two patients (4%) died due to respiratory failure and cardiovascular instability within the first 30 days post-surgery. The 1-year mortality rate was 17% (*n* = 8), but these deaths were not related to flap surgery. At the time of discharge, all successfully treated patients presented with stable soft-tissue conditions and without a sign of recurrent sternal infection.

### 3.2. Follow-Up Examinations

Follow-up donor-site examinations were performed 34 ± 19 months after free flap surgery, including a total of 28 patients (61% follow-up rate). From the initial cohort of 46 patients, 8 patients (17%) had died, and 10 patients (22%) could not be reached. At the TFL donor-sites, the ROM of hip and knee joints revealed no restrictions when compared to the contralateral healthy sides (Table 4). Three patients subjectively described a weakness in knee extension (*n* = 3; 11%); however, muscle strength was not notably impaired in any patient (donor-site: median of 5, range of 4 to 5 vs. healthy-site: median of 5, range of 4 to 5; *p* = 0.25). Herniation of the quadriceps muscle was not seen in any patient at follow-up. The patient-reported satisfaction showed an overall good result for both functional and aesthetic outcomes at the donor-site (function: median of 2, range of 1 to 4; aesthetic: median of 2, range of 1 to 4). With respect to general health and satisfaction, the SF-36 questionnaire (physical component summary, median of 44, range: 33 to 57; mental component summary, median of 29, range: 19 to 40) as well as the LEFS (median of 54, range: 35 to 65) revealed satisfactory results. The mean donor-site scar length was 24.8 ± 3.3 cm. Scar examinations revealed a median VSS score of 3 (range: 2 to 7) at the donor-site and a median of 3 (range: 2 to 5) at the corresponding recipient-site. In accordance, the POSAS showed an overall good satisfaction with scar quality and scar appearance at the donor-site (median: 24, range: 18 to 34) and recipient-site (median: 23, range: 19 to 31) with comparable results for the patients’ and observers’ scale (Table 4).

## 4. Discussion

In the present study, we report on our treatment of a selective group of patients with DSWI and sternal osteomyelitis after cardiac surgery resulting in large sternal defects. The patients in this study had multiple previous surgeries (El Oakley III to V), with their overall morbidity leading to a median ASA score of 3. Nowadays, microsurgical free flap transfer is a safe and reliable procedure, with failure rates ranging between 1 and 6% [17,18,19]. However, it is technically complex, time-consuming, and often debilitating on multimorbid patients due to their diminished physical reserves [20,21,22]. A particularly challenging situation arises when free flaps become the only remaining reconstructive option because defects cannot be closed with local or pedicled flaps. Nevertheless, a persisting disfiguring, painful, and infected defect is hardly ever an alternative for the patient‘s remaining life span. In order to avoid additional complications, any reconstructive procedure in these patients needs to be as safe and reliable as possible. Hereby, operative time should be kept as short as possible, with the designated flap being “easily accessible”. Pursuing the objective of optimizing the treatment of these critically ill patients, we have increasingly been choosing the free TFL flap with its large skin paddle, which makes it ideal for the reconstruction of extensive three-dimensional sternal defects.

The TFL muscle is a weak flexor and lateral rotator of the thigh. The muscle is dispensable, and its absence usually causes no remarkable functional deficit or donor-site morbidity [9]. This was confirmed by the results of our clinical follow-up examinations. Hereby, muscle strength and range of motion were not considerably impaired. However, our method of muscle strength assessment may not have been precise enough to discriminate between the donor-site and uninjured side. The LEFS, a well-established and sensitive tool to measure lower extremity functional impairment [14], also did not reveal considerable impairment in any patient (median LEFS = 54) when compared to normative median values (LEFS = 66) for patients older than 65 [23]. Donor-site morbidity in this study was negligible when compared to other flaps, such as the rectus abdominis muscle flap, for example [24,25]. The TFL flap can comprise a maximal skin paddle three times the size of the muscle [26]. Therefore, it combines the freedom of abundant skin coverage and a strong fascial layer with the versatility of a microvascular pedicle. The fascia is a highly vascularized semirigid layer, which provides additional structural support to cover large defects [9,27]. The TFL flap is easily accessible and can be harvested in a supine position, thus eliminating the need for lengthy intraoperative repositioning, when compared to flaps from the thoracodorsal system. The operating time can be kept short by working in a two-team approach. Another advantage of the large muscular TFL flap is its suitability for anastomoses to AVLs, as opposed to fasciocutaneous perforator flaps, such as the anterior lateral thigh (ALT) flap. In this context, Henn and colleagues stated that ALT flaps in combination with AVLs might be prone to microvascular complications due to an elevated flow resistance of the small-caliber perforator in comparison to the low-resistance conditions of the vein graft used in AVLs [28]. Because of these findings, we refrain from using fasciocutaneous perforator flaps in combination with AVLs and only use muscle-based flaps in these scenarios. In line with this notion, the combination of TFL flaps with AVLs showed no increased risk of microvascular thrombosis in this study.

Certainly, we want to point out that, in general, the full armamentarium of reconstructive procedures is required to cover sternal defects. Hereby, the most prevalent reconstructive options usually comprise pedicled muscle flaps, as they provide well-vascularized tissue with enough bulk to fill the defect cavity. The pedicled VRAM flap, LD flap, and bilateral pectoralis major flap have been the method of choice for decades [5,6,8]. In this context, it is recommended to cover cranial sternal wounds with pectoralis major flaps, whereas VRAM flaps are of better use to cover caudal sternal wounds. Davison and colleagues compared the outcome of 41 modified pectoralis major flaps against 56 pedicled VRAM flaps and reported equal wound dehiscence rates (VRAM: 14.2% vs. PM: 14.6%) [4]. However, it has to be considered that in the majority of patients developing DSWI after cardiovascular surgery, the LIMA is harvested for CABG. Therefore only the RIMA can be used for a cranially pedicled VRAM flap [8]. Furthermore, the most distal part of the pedicled flap, which is of paramount importance to the reconstructive outcome, especially in larger defects, is at risk of impaired perfusion, with a higher risk for wound dehiscence and infection [29]. We agree with Davison and colleagues that both pedicled flaps are straight forward and easy to harvest; nevertheless, the free TFL flap showed lower rates of wound dehiscence and partial flap necrosis in this study.

The latissimus dorsi muscle is the largest human muscle and is ideal to cover larger sternal defects. Even when the insertion to the humeral bone is detached, the maximum arc of rotation of a pedicled LD flap is an important limitation, particularly when a midline exceeding defect must be reconstructed [29]. This can put the most distal part of the flap at risk of impaired perfusion. Spindler and colleagues published a study of 106 patients undergoing sternal reconstruction with a pedicled myocutaneous LD flap. Besides no total flap loss, they reported 35% revision surgeries because of wound healing disorders, hematoma, or persistent infection [30]. In comparison to their results, the number of revision surgeries for partial flap necrosis (*n* = 3), wound healing disorders (*n* = 3), and hematomas (*n* = 5) was lower in our cohort (24%). According to the current literature, the free LD transfer can show reliable results, with encouragingly low rates of revision surgeries or serious complications [31]. However, raising a free LD flap to cover sternal defects has some disadvantages that need to be considered. First, patients must be repositioned intraoperatively, and the flap must be banked in the axilla. Second, the latissimus dorsi muscle, as the rectus abdominis muscle, is an auxiliary breathing muscle; thus, sacrificing this muscle can affect breathing mechanisms in these already multimorbid patients [30]. While some authors state that the greater omentum (OM) flap is a valuable alternative to muscle flaps, we do not consider the OM flap as a first line procedure [32]. Kolbenschlag and colleagues presented a study of 50 patients undergoing sternal defect reconstruction with a pedicled OM flap and reported high surgical complication rates and high donor-site morbidity, with one patient even developing acute intestinal incarceration [33]. Therefore, the OM flap should be considered a last resort backup option rather than a first-line treatment. Furthermore it should be considered than an increasing defect size can be related to a higher incidence of partial flap necrosis of pedicled flaps, leading to a higher rate of revisional surgeries and impaired postoperative recovery [34]. In this context we can recommend the myocutaneous TFL flap as a workhorse flap for extensive sternal defect reconstruction.

Despite the promising nature of our data, which highlight the feasibility of the free TFL flap for the reconstruction of large sternal defects in a one-stage procedure, our study has important limitations that need to be discussed. First, our study is limited by its retrospective nature and involvement of several different surgeons and their various flap planning routines. Second, the study comprises a relatively small number of patients. Therefore, we conclude that a larger cohort of patients and a longer follow-up period are necessary to gain more reliable data regarding the choice of the reconstructive approach in this context. Third, the follow-up response rate was low, and follow-up times varied greatly, which may have resulted in different stages of rehabilitation, wound healing, and scarring assessed.

## 5. Conclusions

In conclusion, the free TFL flap represents a valuable option for sternal reconstruction in critically ill patients with large defects and a history of previously failed reconstructive procedures. It is associated with an encouragingly low morbidity at the corresponding donor-site. We therefore regard the free TFL flap, in combination with AVLs if needed, as a workhorse flap for sternal reconstruction, rather than merely a backup option.

## Figures and Tables

**Figure 1 jpm-12-00427-f001:**
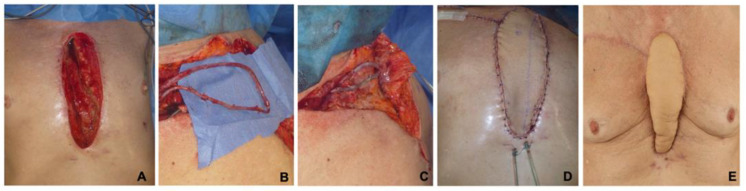
A 62-year-old male patient with DSWI and sternal osteomyelitis after a bilateral CABG procedure. The resulting defect measured 22 × 8 cm (**A**). Sternal reconstruction with a TFL flap and an AVL was planned. The AVL was created between the subclavian artery and vein in an end-to-side technique using a greater saphenous vein graft (**B**). Subsequently, arterial and venous end-to-end anastomoses with the TFL flap pedicle were performed (**C**,**D**). The patient’s recovery was uneventful, and he was discharged 13 days postoperatively. Stable defect reconstruction without any sign of wound healing disorder or recurrent infection three month after surgery (**E**).

**Table 1 jpm-12-00427-t001:** Patient demographics and comorbidities, El Oakley and Wright classification, and microbiological examination (SD = Standard Deviation).

Patient Demographics	
Number of patients and flaps	46
Mean age (years) ± SD (range)	67 ± 11 (38 to 85)
Median ASA class (range)	3 (2 to 4)
Sex (female/male)	17/29
Comorbidities	*n* (%)
Arterial hypertension (HTN)	44 (96%)
Coronary artery disease (CAD)	41 (89%)
Chronic heart failure (CHF)	27 (59%)
Chronic obstructive pulmonary disease (COPD)	20 (44%)
Chronic kidney disease (CKD)	25 (54%)
Diabetes mellitus (DM)	30 (65%)
Active smoker at time of surgery	13 (28%)
BMI (Body Mass Index) (kg/m^2^)	29 ± 6
Obesity (BMI ≥ 30 kg/m^2^)	19 (41%)
El Oakley and Wright classification	
I	-
II	-
IIIA	7 (15%)
IIIB	11 (24%)
IVA	2 (4%)
IVB	-
V	26 (57%)
Microbiological examination of soft and bony tissue	
*Staphylococcus aureus*	17 (37%)
Methicillin-resistant *Staphylococcus aureus*	6 (13%)
*Staphylococcus epidermidis*	14 (30%)
*Enterococcus faecalis*	10 (22%)
*Escherichia coli*	7 (15%)
Multiresistant Gram-negative bacteria	9 (20%)

**Table 2 jpm-12-00427-t002:** Operative characteristics (cm = centimeter, cm^2^ = square-centimeter, min = minute, RIMA = Right Internal Mammary Artery, OT = Operation Time).

Operative Characteristics	
Mean sternal defect size [cm^2^] ± SD	194 ± 43 (128 to 297)
Mean sternal defect length [cm] ± SD	23 ± 3 (18 to 27)
Mean sternal defect width [cm] ± SD	8 ± 1 (7 to 11)
Mean length of flap ischemia [min] ± SD	63 ± 16 (32 to 91)
Mean skin paddle surface [cm^2^] ± SD	205 ± 38 (154 to 308)
Mean flap length [cm] ± SD	24 ± 3 (19 to 28)
Mean flap width [cm] ± SD	8 ± 1 (7 to 11)
Mean OT [min] ± SD (range)	387 ± 120 (212 to 695)
Recipient vessel situation	
RIMA and concomitant vein	9 (20%)
RIMA and cephalic vein	15 (33%)
Cephalic vein-thoracoacromial artery arterio-venous loop	3 (7%)
Cephalic vein-subclavian artery arterio-venous loop	10 (22%)
Subclavian artery/vein arterio-venous loop using a greater saphenous vein graft	9 (20%)

**Table 3 jpm-12-00427-t003:** Postoperative complications.

Surgical Complications	*n* (%)
**Flap Complications**	
Arterial thrombosis	2 (4%)
Venous thrombosis	1 (2%)
Hematoma	5 (11%)
Wound dehiscence	3 (7%)
Partial flap necrosis (>5% of the skin paddle)	3 (7%)
Donor-site complications	
Impaired wound healing	3 (7%)
Wound infection	1 (2%)
Hematoma	1 (2%)

**Table 4 jpm-12-00427-t004:** Follow-up examination (VSS = Vancouver Scar Scale, POSAS = Patient and Observer Scar Assessment Scale, SD = Standard Deviation).

Follow-Up Examinations			
**Range of Motion**	**Donor-Site**	**Healthy Side**	***p*-Value**
Mean knee joint extension/flexion (mean ± SD)	110° ± 9°	114° ± 9°	*p* = 0.08
Mean hip joint extension/flexion (mean ± SD)	118° ± 10°	122° ± 8°	*p* = 0.73
Mean hip joint abduction/adduction (mean ± SD)	69° ± 5°	71° ± 6°	*p* = 0.29
Mean hip internal-/external rotation (mean ± SD)	58° ± 7°	62° ± 7°	*p* = 0.07
**Scarring**	**Donor-Site**	**Recipient-Site**	
Median VSS (range)	3 (2 to 7)	3 (2 to 5)	-
Median POSAS (range)	24 (18 to 34)	23 (19 to 31)	-
Median patients’ scar assessment (range)	12 (9 to 21)	12 (10 to 16)	-
Median observers’ scar assessment (range)	11 (9 to 17)	11 (9 to 15)	-

## Data Availability

The datasets used and/or analyzed during the current study are available from the corresponding author on reasonable request.

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
