# Peer review of "The Free Myocutaneous Tensor Fasciae Latae Flap—A Workhorse Flap for Sternal Defect Reconstruction: A Single-Center Experience"

_jpm, 2022, doi:10.3390/jpm12030427_

Round 1

Reviewer 1 Report

The authors aimed at evaluating the feasibility of this free flap for sternal reconstruction by analyzing operative data, surgical complication rates, reconstructive outcomes, and donor-site morbidity. The paper is well written and well structured. Results are relevant. I support for publication in the current format.

Reviewer 2 Report

The Free Muyocutaneous TFL flap – A Workhorse Flap for Sternal Defect Reconstruction: Single-Center Study.

Sternal reconstruction is a debilitating state, especially in patients with comorbidities. The reconstruction possibilities in this region are abundant, like local flaps, perforator flaps, and pedicled Flaps. In my opinion, one of the best options to reconstruct is to use freestyle perforator flaps, but when the patient doesn’t have IMA due to cardiac surgery, and the defect is of the larger area the only option is a free flap.

The free MTF flap is a very good choice, due to its durability and thickness to cover deep wounds.

I find this study interesting well organized, statistically well prepared, although this is a retrospective study. The described patient groups are well defined. Despite the low volume of the patents, the study presents a good and versatile option of reconstruction.

The other strength of their study is a long follow-up, even the response rate was quite low.

I have only 2 questions:

  1. How long was the heparin prescribed postoperatively?
  2. Did you use any intraoperative flap perfusion devices e.g., ICG, etc.?

It is recommended to continue the prospective study and a present larger group of cases and would be interesting to compare the outcomes between the different clinical centers and populations, but it shows the direction we should follow.

I recommend this manuscript be published as an original article.
